# The Role of RASGRP2 in Vascular Endothelial Cells—A Mini Review

**DOI:** 10.3390/ijms222011129

**Published:** 2021-10-15

**Authors:** Jun-ichi Takino, Shouhei Miyazaki, Kentaro Nagamine, Takamitsu Hori

**Affiliations:** 1Faculty of Pharmaceutical Sciences, Hiroshima International University, 5-1-1 Hirokoshingai, Kure, Hiroshima 737-0112, Japan; s-miyaza@hirokoku-u.ac.jp (S.M.); hori@hirokoku-u.ac.jp (T.H.); 2Faculty of Health Sciences, Hiroshima International University, 5-1-1 Hirokoshingai, Kure, Hiroshima 737-0112, Japan; k-nagami@hirokoku-u.ac.jp

**Keywords:** RAS guanyl nucleotide-releasing protein 2 (RASGRP2), vascular endothelial cells, small GTPase, RAP1, R-RAS

## Abstract

RAS guanyl nucleotide-releasing proteins (RASGRPs) are important proteins that act as guanine nucleotide exchange factors, which activate small GTPases and function as molecular switches for intracellular signals. The RASGRP family is composed of RASGRP1–4 proteins and activates the small GTPases, RAS and RAP. Among them, RASGRP2 has different characteristics from other RASGRPs in that it targets small GTPases and its localizations are different. Many studies related to RASGRP2 have been reported in cells of the blood cell lineage. Furthermore, RASGRP2 has also been reported to be associated with Huntington’s disease, tumors, and rheumatoid arthritis. In addition, we also recently reported RASGRP2 expression in vascular endothelial cells, and clarified the involvement of xenopus Rasgrp2 in the vasculogenesis process and multiple signaling pathways of RASGRP2 in human vascular endothelial cells with stable expression of RASGRP2. Therefore, this article outlines the existing knowledge of RASGRP2 and focuses on its expression and role in vascular endothelial cells, and suggests that RASGRP2 functions as a protective factor for maintaining healthy blood vessels.

## 1. RAS Guanyl Nucleotide-Releasing Protein (RASGRP) Family

Small GTPases that function as molecular switches and transmit intracellular signals become active by replacing bound GDP with GTP. This GDP to GTP replacement is stimulated by guanine nucleotide exchange factor (GEF), which competes with GTPase-activating proteins that catalyze GTP hydrolysis [1]. The RASGRP family, which acts as a GEF and is composed of four members including RASGRP1–4, activates the RAS and RAP proteins of small GTPases. All RASGRPs have a RAS exchange motif (REM), a CDC25 domain, which is a catalytic GEF domain, an EF hand that is a calcium-binding domain, and a C1 domain, which is a diacylglycerol (DAG)-binding domain (Figure 1) [2]. The C1 domain of RASGRP proteins allows translocation to membranes by binding to DAG or its analogs [3]. In addition, RASGRP1 and RASGRP3 have a plasma membrane-targeting domain involved in membrane translocation, and it has been reported that these domains are controlled or affected by EF1, one of the EF hands of RASGRP1 and the REM region of RASGRP3, respectively [4,5]. RASGRP1 and RASGRP4 activate RAS, while RASGRP3 activates RAP in addition to RAS [6].

## 2. RASGRP2 Characteristics

Among the RASGRP family members, RASGRP2, also called CALDAG-GEFI, activates RAP1 [6]. However, a study using 293T cells expressing both RASGRP2 and each small GTPase individually by transfected vectors showed that RASGRP2 slightly activates R-RAS, TC21, and RAP2 [7,8]. RAP1 and RAP2 belong to the RAP subfamily, and RAP1 in vascular endothelial cells is also activated by other GEFs, such as EPAC, PDZ-GEF, and C3G [9]. R-RAS and TC21 belong to the R-RAS subfamily; however, their functions are known to be different from the classic RAS proteins of the RAS family [10]. In particular, R-RAS has been reported to be highly expressed in the vascular endothelial cells [11].

Unlike other members, RASGRP2 cannot bind to the DAG and translocate to the membrane because the loop A and loop B parts of the C1 domain are mutated [3]. However, it translocates by binding to F-actin via the REM region [12]. The GEF activity of RASGRP2 is regulated by the release of autoinhibition associated with the rearrangement of the EF hands caused by calcium binding [13], and is suppressed by intracellular calcium chelators such as BAPTA-AM [14]. On the other hand, it has also been clarified that the GEF activity can be regulated by phosphorylation, which is a post-translational modification. In a study using HEK293 cells expressing RASGRP2 or its amino acid substitution mutants, it was shown that protein kinase A (PKA) phosphorylates Ser116/Ser117 (weakly) and Ser587 (strongly) of RASGRP2 [15]. In particular, the phosphorylation of RASGRP2 at Ser587 almost attenuated the activation of RAP1 in transfected HEK293 cells and human platelets [15]. Furthermore, in human platelets, RASGRP2 phosphorylation has also been reported to be downregulated by P2Y12 receptor-mediated ADP stimulation and upregulated by the prostacyclin derivative, Iloprost [16]. Phosphoproteomic analysis of acute slices of mouse striatum confirmed that the dopamine type 1 receptor agonist, SKF81297, and the PKA activator, forskolin, phosphorylate Ser116/Ser117, Ser554, and Ser586 of mouse RASGRP2 [17]. In addition, in transfected HEK293T cells, phosphorylation of Ser394 in the linker region results in the autoinhibition of RASGRP2 by BRaf-MEK-ERK signaling, and reduces its GEF activity [18].

Several studies on the function of RASGRP2 and its mutation or deficiency have been reported in human platelets and leukocytes [19,20,21,22,23]. RASGRP2 is associated with immune-mediated thrombosis and thrombocytopenia [20], integrin-independent neutrophil chemotaxis [21], and regulation of platelet and T-cell adhesion via integrin [22,23]. In contrast, RASGRP2 has been reported to be associated with the striatum of mouse Huntington’s disease models [24], prostate cancer [25], fibroblast-like synoviocytes from rheumatoid arthritis synovium [26], and osteosarcoma [27].

## 3. Other RASGRPs in Vascular Endothelial Cells

Although few studies of RASGRPs in vascular endothelial cells have been reported, endogenous RASGRP1 knockdown using siRNA in human umbilical vein endothelial cells (HUVECs) decreased vascular endothelial growth factor (VEGF)-induced migration and tube formation. In addition, RASGRP1 has been shown to be required for AKT phosphorylation in the VEGF signaling pathway, and is a VEGF-responsive gene. It has also been reported that RASGRP1 is involved in the pro-angiogenic effects of the anti-diabetic drug, metformin, under hyperglycemia [28].

*Rasgrp3* has been identified as a novel vascular gene expressed in vascular endothelial cells using an embryonic stem (ES) cell-based gene trap screen. In addition, RASGRP3 is expressed during vascular development and downregulated in mature adult blood vessels, but is expressed in newly formed blood vessels during pregnancy and tumorigenesis. When wild-type blood vessels derived from ES cells were exposed to phorbol myristate acetate, a DAG analog, abnormal morphogenesis of the vascular endothelium, depending on the presence of RASGRP3, was observed [29]. On the other hand, it has also been shown that loss of RASGRP3 function does not affect survival. Considering the expression of *Rasgrp2*, which was detected by RT-PCR analysis using a murine endothelial cell line, it has been suggested that RASGRP2 function may compensate for the loss of RASGRP3 [29]. RASGRP3 expression in HUVECs, similar to that of RASGRP1, is upregulated by VEGF, and RASGRP3 signaling via the RAS/MEK/ERK pathway by endothelin-1 affects the endothelial actin cytoskeleton and perturbs migration [30]. Therefore, it has also been suggested that RASGRP3 exacerbates diabetic vascular complications characterized by excess DAG or causes diabetes-induced developmental disorders. To date, there have been no reports on RASGRP4 related to vascular endothelial cells.

## 4. Rasgrp2 Expression in Vasculogenesis during Xenopus Development

Vasculogenesis is the process of vascular development that begins with the formation of angioblasts from the mesoderm and their subsequent migration, mediated by vascular endothelial growth factor signaling [31,32].

The expression of xenopus *rasgrp2* (*xrasgrp2*; homolog of the human *RASGRP2*) mRNA in vascular endothelial cells was reported in Xenopus embryos [33]. The expression of 143 transcripts, including that of vascular-expressed genes, was revealed by microarray analysis using the constructed aggregates from xenopus animal cap cells that were co-treated with activin and angiopoietin-2, which expressed the vascular endothelial markers *X-msr*, *Xtie2*, and *xegfl7* [34,35]. Furthermore, *xrasgrp2* was identified as a novel vascular-expressed gene by expression pattern analysis [33]. The temporal expression pattern of *xrasgrp2* mRNA in xenopus embryos was observed after stage 24, and the timing was consistent with that of vascular development. The spatial expression pattern of *xrasgrp2* mRNA in xenopus embryos was observed in vascular regions such as the eye, intersomitic vein, posterior cardinal vein, aortic arch, and vascular vitelline network. In addition, xenopus Rasgrp2 overexpression induced ectopic vascular formation and its knockdown delayed vascular development. VEGF-induced Rasgrp2 expression caused VEGF-induced hemangioblast cell-to-vascular endothelial cell differentiation by expressing *X-msr* and *Xtie2* while suppressing *globin T3* [36]. Therefore, it has been suggested that Rasgrp2 is essential in the early phase of vasculogenesis during xenopus development (Figure 2).

## 5. *RASGRP2* Expression in the Human Vascular Endothelial Cells

The *RASGRP2* expression in humans has been demonstrated by RT-PCR screening using human umbilical artery endothelial cells (HUAECs) and the HUVECs. The human *RASGRP2* gene has three alternative variants due to differences in the first non-coding exons, and HUAECs express *RASGRP2* mRNA containing the first exon distal to the 5’-untranslated region. In addition, regulation of *RASGRP2* expression has also been shown by luciferase and gel super shift assays: The promoter and silencer regions are upstream of the distal first exon, and the octamer-binding transcription factor 1 binds to the silencer region [37].

## 6. Suppression of Apoptosis by RASGRP2 Signaling in the Vascular Endothelial Cells

Apoptosis of vascular endothelial cells is caused by various factors, such as hyperglycemia and inflammation. Apoptosis induces tissue damage and affects the onset and exacerbation of disease [38,39,40,41,42,43]. On the other hand, it is also related to remodeling into a mature network during angiogenesis [44]. Therefore, vascular endothelial cell apoptosis is a very important event in various diseases and angiogenesis. The apoptotic pathway is roughly divided into the mitochondrial pathway, which is an intrinsic pathway, and the death receptor pathway, which is an extrinsic pathway. The mitochondrial pathway is induced in response to cellular stress and results in the activation of pro-apoptotic BH3-only proteins, such as BIM. Subsequently, BIM directly or indirectly activates BAX and leads to apoptosis via the activation of caspase-9. On the other hand, the death receptor pathway is induced by interacting with the death receptors, which are cell surface receptors, and their ligands, such as tumor necrosis factor-α (TNF-α), and results in apoptosis via activation of caspase-8 [45].

An analysis using human telomerase reverse transcriptase (hTERT)-immortalized HUVECs (TERT HUVECs) with stable overexpression of RASGRP2 initially revealed that RASGRP2 activates RAP1A, R-RAS, and TC21 but not RAP2A in human vascular endothelial cells [46]. Furthermore, RAP1 activated by RASGRP2 suppresses apoptosis through TNF-α, and the suppression mechanism is due to the inhibition of reactive oxygen species (ROS) production mediated by NADPH oxidase (NOX) [47]. Furthermore, R-RAS activated by RASGRP2 suppressed apoptosis through BAM7 or anisomycin, which are BAX activators, and the suppression was due to the inhibition of BAX translocation by promoting hexokinase-2 translocation to mitochondria via R-RAS-PI3K-AKT signaling pathway. In addition, in a siRNA knockdown of endogenous RASGRP2 in TERT HUVECs, which express similar levels of RASGRP2 as HUVECs, the reduction of RASGRP2 expression decreased steady-state R-RAS activity and increased BAM7-induced apoptosis [46]. Therefore, it has been suggested that RASGRP2 plays an important role in cell survival (Figure 3).

## 7. Suppression of Vascular Hyper-Permeability by RASGRP2 Signaling in Vascular Endothelial Cells

Vascular hyperpermeability caused by various factors, such as advanced glycation end products (AGEs) and lipopolysaccharides, affects the onset and exacerbation of disease [48,49,50,51]. Maintenance of vascular permeability is regulated by adhesion factors between vascular endothelial cells, including adherens junctions (AJs), tight junctions (TJs), and gap junctions, which consist of vascular endothelial (VE)-cadherin and its associated α-, β-, and p120-catenin adhesion complexes; and occludin; claudins; junctional adhesion molecules; and associated zonula occludens (ZO)-1, -2, and -3 proteins [52].

RAP1 activated by RASGRP2 suppressed vascular hyperpermeability through the NOX-ROS pathway by AGEs without enhancing the basal barrier function. R-RAS activated by RASGRP2 partially suppressed vascular hyperpermeability through the non-ROS producing pathway by AGEs. The suppression by RASGRP2 was ultimately due to protection against perturbation of the VE-cadherin protein, but not the ZO-1 protein [53]. Therefore, it has been suggested that RASGRP2 plays an important role in vascular permeability (Figure 4).

## 8. Conclusions

Although XRASGRP2 was identified during the vasculogenesis process of xenopus development, changes in the expression of RASGRP2 during vasculogenesis and angiogenesis in humans have not yet been studied. In vascular endothelial cells, RAP1 plays an important role in cell–matrix and cell–cell adhesion, migration, and tube formation, that is, it is required for normal vasculogenesis and angiogenesis [54]. R-RAS promotes lumenogenesis, which is fundamental to angiogenesis by stabilizing the microtubule cytoskeleton [55]. Furthermore, it contributes to the maturation of blood vessels by stabilizing the endothelial barrier and interacting with pericytes [56,57]. Since these small GTPases play an important role in angiogenesis, the relationship between their function and RASGRP2 requires further investigation.

On the other hand, in a study using human vascular endothelial cells, it was found that RASGRP2 activates not only RAP1 but also R-RAS and induces multiple signaling pathways. It has been suggested that RASGRP2 is involved in the suppression of apoptosis and hyper-permeability by activating small GTPases, such as RAP1 and R-RAS, in vascular endothelial cells. Therefore, RASGRP2 in vascular endothelial cells may act as a protective factor to maintain healthy blood vessels.

## Figures and Tables

**Figure 1 ijms-22-11129-f001:**
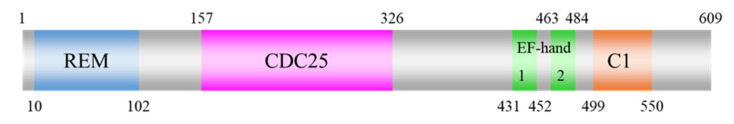
Domain architecture of RASGRP2. RASGRP2 is a multidomain protein consisting of a REM region, a CDC25 domain, two EF hands, and a C1 domain. GenomeNet server (https://www.genome.jp/tools/motif/, accessed on 14 October 2021) was used to search for specific motifs in protein sequences.

**Figure 2 ijms-22-11129-f002:**
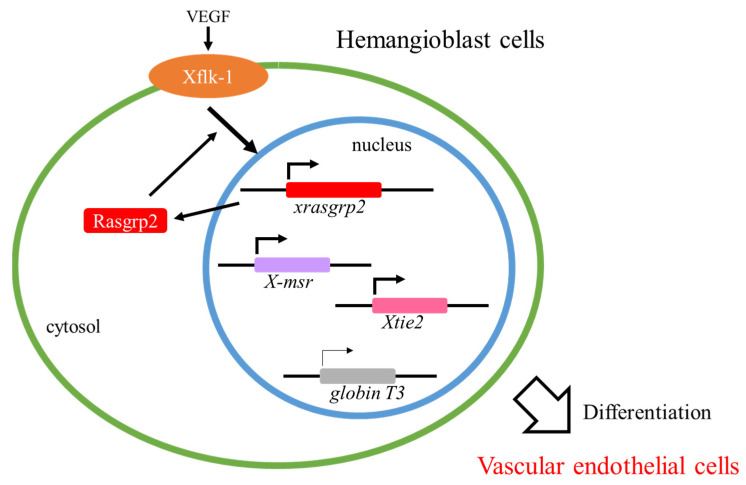
Proposed model for Rasgrp2 function in hemangioblast cells. VEGF: vascular endothelial growth factor, Rasgrp2: Ras guanyl nucleotide-releasing protein 2.

**Figure 3 ijms-22-11129-f003:**
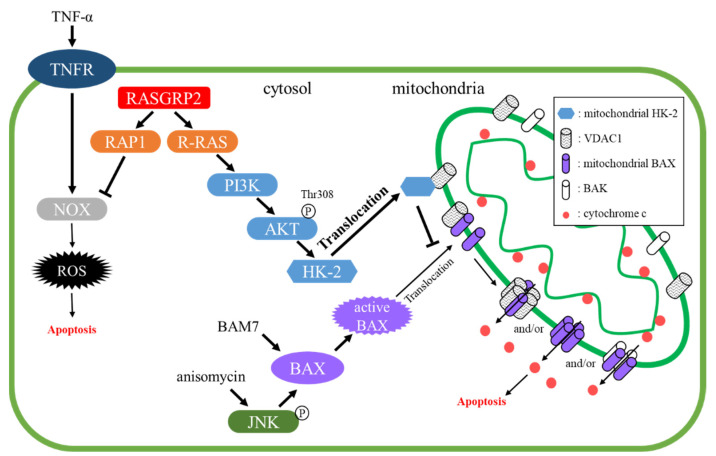
Proposed model for apoptosis suppression via the RAP1 and R-RAS pathways by RASGRP2. TNF-α: tumor necrosis factor-α, TNFR: tumor necrosis factor receptor, NOX: NADPH oxidase, ROS: reactive oxygen species, RASGRP2: RAS guanyl nucleotide releasing protein 2, PI3K: phosphoinositide 3-kinase, JNK: c-jun N-terminal kinase, HK-2: hexokinase-2, VDAC: voltage-dependent anion channel, Thr: Threonine, P: phosphorylation.

**Figure 4 ijms-22-11129-f004:**
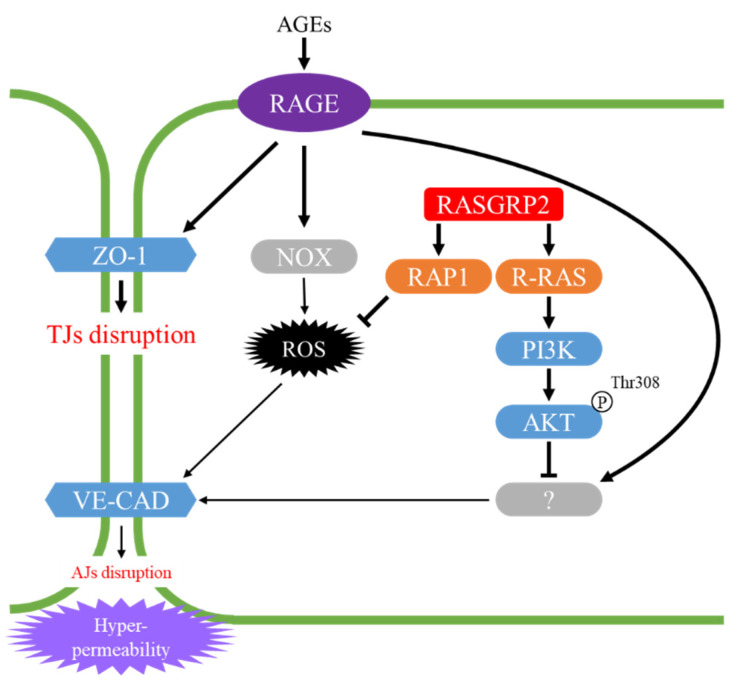
Proposed model for vascular hyper-permeability suppression via the RAP1 and R-RAS pathways by RASGRP2. AGEs: advanced glycation end products, RAGE: receptor for AGEs, ZO-1: zonula occludens-1, VE-CAD: vascular endothelial-cadherin, TJs: tight junctions, AJs: adherens junctions.

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
