# Peer review of "The Role of RASGRP2 in Vascular Endothelial Cells—A Mini Review"

_ijms, 2021, doi:10.3390/ijms222011129_

Round 1

Reviewer 1 Report

In this paper the authors review the RasGRPs structure as multidomain proteins sharing common structural organization and in particular RasGRP2  that diverges from the other members of the RasGRP family as it catalyzes GDP to GTP exchange only for Rap GTPases but not Ras.

The authors argue about the  RasGRP2 as protective factor for vascular endothelial cells.

 COMMENTS

In my opinion the paper is well written and gives a good overview of the issue.

Author Response

Reviewer 1

In my opinion the paper is well written and gives a good overview of the issue.

-Thank you for your comment on this paper.

Reviewer 2 Report

Major points:

Title: Having looked at the main text, the ‘protective’ role of RasGRP2 in vascular endothelial cells is not well defined or established, although there are studies showing RasGRP2 in suppression of apoptosis and vascular hyperpermeability in vascular endothelial cells. The authors are encouraged to have the title revised.

Line 83: for this section, instead of listing the individual characteristics of other RasGRPs, the authors are encouraged to develop this section in a more comparative manner with comparison to RasGRP2.

Line 132 – 139: This section needs to be expanded to accommodate the subtitle which claims to discuss the regulation RasGRP2 in human. Otherwise, please revise the subtitle.

Minor points:

Please use the standard gene nomenclature throughout the text. For example, in Line 133, it is the gene symbol for human nomenclature, and thus should be capitalized.

The language requires further editing. For instance, in the abstract, the authors wrote ‘in contrast, RasGRP2 has also been …’when there’s no divergence or comparison in the context.

Author Response

Reviewer 2

-Thank you for your comments on this paper. We totally agree with the points raised by the reviewers and amended the text considering their suggestions. The revised parts within the manuscript are indicated in red.

  • Title: Having looked at the main text, the ‘protective’ role of RASGRP2 in vascular endothelial cells is not well defined or established, although there are studies showing RASGRP2 in suppression of apoptosis and vascular hyperpermeability in vascular endothelial cells. The authors are encouraged to have the title revised.

-Thank you for the suggestion. We agree with the reviewer’s comment. In the revised manuscript, we changed the title (L2).

  • Line 83: for this section, instead of listing the individual characteristics of other RASGRPs, the authors are encouraged to develop this section in a more comparative manner with comparison to RASGRP2.

-We thank the reviewer for this suggestion. We think that studies of other RASGRPs in vascular endothelial cells are very important but only a few have been reported and are difficult to compare with studies of RASGRP2.

Therefore, in this manuscript, we would like to mention in the section that distinguishes the roles of RASGRP2 and other RASGRPs in the vascular endothelial cells.

  • Line 132 – 139: This section needs to be expanded to accommodate the subtitle which claims to discuss the regulation RASGRP2 in human. Otherwise, please revise the subtitle.

-Thank you for the suggestion. We agree with the reviewer’s comment. In the revised manuscript, we changed the subtitle (L132).

Minor points:

  • Please use the standard gene nomenclature throughout the text. For example, in Line 133, it is the gene symbol for human nomenclature, and thus should be capitalized.

- We thank the reviewer for this suggestion. In the revised manuscript, we have corrected all gene and protein names according to the nomenclature

  • The language requires further editing. For instance, in the abstract, the authors wrote ‘in contrast, RASGRP2 has also been …’when there’s no divergence or comparison in the context.

- We thank the reviewer for this suggestion. In the revised manuscript, we corrected the point out part.

Round 2

Reviewer 2 Report

The authors have addressed my comments.